# From Gut to Blood: Spatial and Temporal Pathobiome Dynamics during Acute Abdominal Murine Sepsis

**DOI:** 10.3390/microorganisms11030627

**Published:** 2023-02-28

**Authors:** Christina Hartwig, Susanne Drechsler, Yevhen Vainshtein, Madeline Maneth, Theresa Schmitt, Monika Ehling-Schulz, Marcin Osuchowski, Kai Sohn

**Affiliations:** 1Innovation Field In-Vitro Diagnostics, Fraunhofer Institute for Interfacial Engineering and Biotechnology IGB, 70569 Stuttgart, Germany; 2Institute for Interfacial Engineering and Plasma Technology IGVP, University of Stuttgart, 70049 Stuttgart, Germany; 3Ludwig Boltzmann Institute for Traumatology the Research Centre in Cooperation with AUVA, 1200 Vienna, Austria; 4Functional Microbiology, Department of Pathobiology, Institute of Microbiology, University of Veterinary Medicine Vienna, 1210 Vienna, Austria

**Keywords:** cecal ligation puncture, CLP, gut microbiome, cell-free DNA, pathogen liquid biopsy, sepsis, pathobiome, next-generation sequencing, NGS

## Abstract

Abdominal sepsis triggers the transition of microorganisms from the gut to the peritoneum and bloodstream. Unfortunately, there is a limitation of methods and biomarkers to reliably study the emergence of pathobiomes and to monitor their respective dynamics. Three-month-old CD-1 female mice underwent cecal ligation and puncture (CLP) to induce abdominal sepsis. Serial and terminal endpoint specimens were collected for fecal, peritoneal lavage, and blood samples within 72 h. Microbial species compositions were determined by NGS of (cell-free) DNA and confirmed by microbiological cultivation. As a result, CLP induced rapid and early changes of gut microbial communities, with a transition of pathogenic species into the peritoneum and blood detected at 24 h post-CLP. NGS was able to identify pathogenic species in a time course-dependent manner in individual mice using cfDNA from as few as 30 microliters of blood. Absolute levels of cfDNA from pathogens changed rapidly during acute sepsis, demonstrating its short half-life. Pathogenic species and genera in CLP mice significantly overlapped with pathobiomes from septic patients. The study demonstrated that pathobiomes serve as reservoirs following CLP for the transition of pathogens into the bloodstream. Due to its short half-life, cfDNA can serve as a precise biomarker for pathogen identification in blood.

## 1. Introduction

A life-threatening organ dysfunction caused by a dysregulated host response to infection defined as sepsis [1] is caused by the influx of microbes into the bloodstream. The successful treatment of systemic infections critically depends on the time of diagnosis that in turn is contingent upon a most specific and sensitive microbiological analysis of the patient’s blood sample. A more precise diagnosis and early pathogen identification is lifesaving for septic patients [2]. However, microbial culture is still conceived as the gold standard for the confirmation of systemic bacteremia, yet it is lengthy and frequently burdened by false-negative results [3,4,5].

Apart from the sheer detection of live pathogen(s) in the blood at admission and/or time of concern, the protracted dynamic of the pathogen fluctuations in the septic patient’s systemic circulation remains poorly understood. The delay of the blood culture, heterogeneity of sepsis presentation, and unsatisfactory performance of various scoring systems in the identification of bacteremia [6] leave a large knowledge gap. This limitation equally pertains to sepsis-causing mono- as well as poly-bacterial infections. A monobacterial infection induces massive rearrangements of the existing microbiome and host’s immunity that frequently trigger the opportunistic and/or gut microbiota to complicate the initially single-pathogen invasion [7,8]. Any polymicrobial infection is not static but instead undergoes rapid fluctuations regarding both its qualitative and quantitative characteristics. Thus, a detailed qualitative/quantitative characterization of the time course, sequence, and spread of the pathobiome in an individual septic subject is desired, as it would likely enhance the efficacy of antimicrobial treatment.

Abdominal sepsis is the most common example of a polymicrobial infection, typically originating from the gastrointestinal compartment of the host upon its damage (e.g., perforated appendicitis, diverticulitis, postsurgical leakage). Polymicrobial infections from the abdominal compartment are typically the second most frequent (after the pulmonary compartment) cause of sepsis [9]. On the experimental level, the cecal ligation and puncture (CLP) mouse model closely replicates the clinical characteristics of human abdominal sepsis and constitutes the most frequently used model of sepsis in laboratory rodents [10]. Approximately 90% of the gut microbiota genera (e.g., *Bacteroides*, *Clostridium*) are shared by both humans and mice [11,12]. Additionally, there are distinct interspecies similarities between dominant gut microbial communities (i.e., enterotypes) [13,14,15] indicative of similar patterns of genera and/or abundance shifts in the gut of mice and humans challenged by the same condition [12,13].

To address the above-defined knowledge gap, we devised a tailor-made analytical workflow for the mouse CLP model to characterize individualized microbial dynamics before and during the acute phase of polymicrobial sepsis. Specifically, we monitored individual septic mice to elucidate qualitative/quantitative fluctuations of their pathobiome across the blood, gut, and abdominal compartments. Finally, we compared for the first time the blood pathobiome between the mouse CLP and human patients with abdominal sepsis. We also present a workflow for cfDNA as a sensitive biomarker for individualized pathogen diagnostics and potential clinical application.

## 2. Materials and Methods

Animals: Twelve-week-old female CD-1 mice were purchased from Charles River Laboratories (Sulzfeld, Germany). Groups of 5 animals were housed in type III cages under standardized conditions (i.e., 12 h light–dark diurnal cycle, controlled temperature of 22–24 °C). A standard rodent diet and fresh water were provided ad libitum throughout the study. Cages were enriched with items for gnawing to facilitate natural rodent behavior.

Sepsis model: Mice underwent cecal ligation and puncture (CLP)-inducing polymicrobial abdominal sepsis. Mice were anesthetized under inhalation of isoflurane (2–3%, Forane^®^, Baxter, Vienna, Austria) and received buprenorphine (Bupaq 0.1 mg/kg; Richter Pharma, Austria) as analgesic control before undergoing midline laparotomy. CLP was performed according to the original protocol [16] with modifications described elsewhere [17]. In brief, a ligature was placed under the ileocecal valve to induce a medium-severity sepsis. The cecum was punctured twice using a 17-gauge needle, and a small amount of fecal content was extruded to ensure patency of the puncture. The abdominal cavity was closed with sutures, and the skin was closed with a wound glue. The CLP group was subdivided into an antibiotic group and a nonantibiotic group, with the former being treated with Imipenem (25 mg/kg with Cilastatin) subcutaneously 2 h after surgery and in 12 h intervals thereafter for 3 days together with buprenorphine [18]. Most of the mice appearing in the main text did not receive antibiotic treatment to enable following the infection with subsequent transition of microbial species without interfering antibiotics. Mice with treatment were not analyzed separately. The control group underwent a sham surgery consisting of a midline laparotomy without ligation and puncture with a brief exteriorization of the cecum. In this study, a total of 43 mice were used for experiments. Thereof, 4 mice underwent sham surgery, 2 being sacrificed at 0 h and 72 h, respectively. Twenty-two CLP mice were sacrificed at 24 h time point, 13 at 48 h time point, and 4 at 72 h time point. All animal experiments were performed in the Ludwig-Boltzmann Institute for Traumatology, Vienna, Austria, under the approval of the local legislative ethical committee (Animal Use Proposal Permission No. 271308/2014/13 and 343130/2013/14) and conducted observing National Institutes of Health guidelines.

Monitoring: All mice were monitored for clinical signs of illness, and their status was evaluated using our custom-developed modified mouse clinical assessment scoring system (M-CASS) based on, for example, fur, posture, mobility, alertness, startle, and righting reflex [19] starting 12 h post-CLP. Simultaneously, rectal temperature was monitored (Fluke Series II thermometer, Fluke, Everett, USA) at least twice daily (or more often whenever a mouse deteriorated) to ensure a maximally precise monitoring of humane endpoints as performed in our previous studies [19,20,21]. Mice were deemed moribund whenever the righting reflex was absent, M-CASS score was ≥8, and/or body temperature (BT) was <28 °C (recorded in at least two sequential measurements) and immediately euthanized under deep-inhalation anesthesia with isoflurane followed by cervical dislocation. BTs can be seen in Appendix A. Body weight was measured daily. All mice in the septic group (24.2–41.5 g) were comparable to sham mice (26.1–35 g).

Study design: Mice were sacrificed at different time points (0–72 h) after surgery to enable a longitudinal follow-up (Figure 1). According to FELASA regulations, each animal was subjected to an individualized sampling pattern (e.g., −24 h, 0 h, 24 h, 48 h, 72 h) to not exceed a critical blood volume threshold per mouse. Hence, an individual mouse was subjected to a serial blood sampling at most 5 times during the study with at least 12 h between sampling time points. Unique mouse IDs and a description of excluded samples can be found in Appendix A. The study was designed to maximally comply with the Minimum Quality Threshold in Preclinical Sepsis Studies (MQTiPSS) consensus guidelines [22].

Sampling: A serial low-volume blood sampling (30 µL, no sacrifice) was performed according to the previously published method [23]. Serial blood sampling enables a repeated sampling in the same mouse, facilitating individual protracted monitoring and reducing the number of mice in the study (3R tenet; reduction). Briefly, an approximate volume of 25–30 µL of blood (depending on the body weight of an individual mouse) was collected with a pipette from the facial vein (vena facialis) punctured with a 23-G needle. Samples were then immediately diluted 1:10 in PBS with ethylenediaminetetraacetic acid (EDTA). After centrifugation (1000× *g*, 5 min, 22 °C), approximately 270 μL of plasma was removed and stored at −80 °C for analysis. Terminal blood samples (0.35–1.0 mL) were collected at sacrifice from vena cava into K3-EDTA tubes. Peritoneal lavage was collected at the sacrificing time point only as described previously [19] based on the original protocol of Ray and Dittel [24]. The pellet of centrifuged lavage was stored at −80 °C. Sampling of feces took place (i) during CLP surgery and (ii) at the time point of sacrifice for each mouse. At CLP, a small amount of fecal content from the puncture sites was extruded by carefully applying pressure to the cecum. At sacrifice, feces were typically collected from the colon in the proximity of the cecum, given the developing necrosis in the ligated cecum. Fecal samples were collected with the needle and subsequently stored at −80 °C.

Human patients and controls: Data from 48 patients with septic shock were taken from a previously published study [25]. The primary septic focus of the 48 patients also shown in this study was the abdomen (*n* = 43; 90%), followed by the lungs (*n* = 4; 8%) and the genitourinary tract (*n* = 1; 2%). The ethics agreement of the Ethics Committee of the Medical Faculty of Heidelberg (Trial Code No. S-097/2013) allowed data collection in the abovementioned study. In this study, we reanalyzed the data already generated.

DNA isolation: Plasma preparation and nucleic acid isolation were performed as described previously [25,26]. Blood plasma was used for automated DNA isolation using the QIAsymphony^®^ SP DNA Preparation System. The QIAsymphony DSP Circulating DNA Kit was used according to the manufacturer’s protocol.

Lavage pellets were thawed on ice and resuspended in 400 μL of yeast-lysis buffer (100 mM NaCl, 10 mM Tris-HCl, 1 mM EDTA, 2% *v*/*v* Triton X-100, 1% *w*/*v* SDS). The resulting suspension was transferred into a reaction tube containing 400 μL of glass beads with a diameter of 0.5 mm. Then, 400 μL phenol/chloroform/isoamyl alcohol in a ratio of 25:24:1 was added, and 3 cycles of 5 min vortexing and a subsequent 5 min incubation on ice were performed. Afterward, samples were centrifuged for 10 min at 17,000× *g* at 4 °C, and the aqueous phase was transferred into a new reaction tube. Next, 1 mL ice-cold 100% ethanol was added to the reaction tube, vortexed thoroughly, and then incubated at −20 °C for 2 h. The sample was again centrifuged for 10 min at 17,000× *g* at 4 °C, and the supernatant was discarded. Subsequently, the pellet was air-dried for 5 min and resuspended in 60 μL of nuclease-free water. Isolated genomic DNA (gDNA) was purified using Agencourt AMPure XP Magnetic Bead purification, applying a ratio of 1.8:1 bead to sample volume.

The Quick-DNA Fecal/Soil Microbe Microprep Kit (Zymo Research; Freiburg, Germany) was used for DNA isolation from fecal samples according to the instruction manual. Isolated gDNA underwent the same bead-based purification step as DNA isolated from lavage samples.

Isolated DNA samples underwent concentration measurements and quality control by the Qubit dsDNA HS Assay Kit (Thermo Fisher Scientific, Waltham, MA, USA) and by the High Sensitivity Genomic DNA Analysis Kit using a fragment analyzer (Agilent, Santa Clara, CA, USA) or an ultrasensitivity NGS analysis kit using Femto Pulse (AGILENT). cfDNA concentrations of terminal blood samples can be seen in Appendix A.

*NGS library preparation and sequencing:* NGS library preparation for human cfDNA samples was performed as previously described [25,26] with 1 ng cfDNA input using the transposase-based Nextera XT library preparation kit (Illumina, San Diego, CA, USA) with a Biomek FX^P^ liquid-handling robot (Beckman Coulter, Brea, CA, USA). The final elution volume was 34 µL of resuspension buffer following the final bead cleanup.

Genomic DNA from murine lavage and fecal samples was prepared manually with the Nextera XT library preparation kit following the manufacturer’s instructions (Illumina, San Diego, CA, USA) and also with 1 ng input in a volume of 5 µL and a final elution volume following the final bead cleanup of 38 µL.

Murine cfDNA from blood plasma samples was prepared by either a Biomek FX^P^ liquid-handling robot using the adapter ligation-based NEXTFLEX Cell Free DNA-Seq Kit with 0.5 ng input and an elution volume of 32 µL or manually using the NEBNext Ultra II DNA library preparation kit. NEXTFLEX was performed with 0.5 ng DNA input at a volume of 32 µL. Adapters were used in a dilution of 1:30, and final elution was performed in 14 µL of nuclease-free water. Libraries showing adapter peaks higher than the library peak on fragment analyzer profiles underwent an additional bead-based purification step in a bead-to-sample ratio of 0.6–0.8:1. The manual preparation of NEBNext Ultra II DNA libraries was done according to the manufacturer’s protocol. DNA input of control mice and sepsis mice was 0.5 ng and 5 ng, respectively. For 0.5 ng DNA input, 10 PCR cycles were performed; for 5 ng DNA input, 8 cycles were performed. Final libraries were eluted in 33 μL of nuclease-free water.

Sequencing of DNA libraries was performed by HiSeq2500 (Illumina), generating 15–25 million single-end reads for murine blood plasma samples and 10 million reads for murine fecal and lavage samples.

Bioinformatic analyses: Bioinformatic analysis was performed as already published with adaptions to murine samples [25,26]. In short, raw reads were separated from potential adapter contamination, quality controlled, and, if necessary, trimmed using BBDuk [27]. Read quality needed to surpass a Phred score of 20 with a minimal length of 50 bp after trimming of low-quality and adapter bases. Subsequently, NextGenMap [28] was used to align quality controlled reads to the murine reference genome M. *musculus*_GRCm38, requiring-minimum identity between reads and a reference genome of 65%. Reads mapping to the murine reference genome and reads with low complexity (consecutive stretches of di- and trinucleotides along the whole read sequence) were excluded from further analysis using prinseq-lite [29]. Finally, Kraken [30] was used to assign reads to systematic classification using the RefSeq database (release version 68).

To be able to compare microbial burdens of different blood samples irrespective of their sampling time point and volume, a calculation was performed to receive an absolute value for the number of DNA molecules of one single microbial species per milliliter of blood:
$$\frac{\mathrm{molecules}}{\mathrm{ml}}=\frac{\mathrm{species reads}}{\mathrm{library size}}*\mathrm{complexity}*\frac{\mathrm{cfDNA concentration}}{M(\mathrm{microbial cfDNA})}*A_{N}$$

The formula for the calculation of molecules/mL includes “species reads” that are all sequenced reads that were classified on the species level. The “library size” represents all quality-trimmed reads that were assigned to one sample according to the barcode. The complexity factor is a measure of the degree of duplication. Another important factor is cfDNA concentration in the blood given as g/mL calculated by blood sampling volume and cfDNA yield after isolation measured by Femto Pulse (smear analysis 50–250 bp) in triplicate. To determine the number of DNA molecules, the molar mass *MM* = 128,700 g/mol of microbial cfDNA with a typical average length of 198 bp and the Avogadro constant *A_N_* = 6.022 ∗ 10^23^ 1/mol were considered. The average length of microbial cfDNA was determined by analyzing a length distribution profile of microbial DNA of *E. coli*, *E. hirae*, and *B. vulgatus*.

Microbiology: Serial dilutions of terminal blood and peritoneal lavage samples were plated in duplicate on Columbia III agar, supplemented with 5% sheep blood (Beckton Dickinson, Heidelberg, Germany). Samples were incubated in parallel aerobically and anaerobically at 37 °C for 24 h. Five colonies per plate were picked and subcultivated on tryptone soy agar plates under the same conditions. Identification of microbial species was performed by matrix-assisted laser desorption and ionization time of flight (MALDI-TOF) as described previously [31].

Cytokine measurements: Cytokines were measured by a highly sensitive automatized Luminex bead-based technology (Bio-Plex Pro™ Mouse Cytokine Th17 Panel A 6-Plex #M6000007NY, Bio-Rad) according to manufacturer’s insctructions.

Software: GraphPad Prism 9.0.0 and BioRender (BioRender.com, accessed on 9 January 2023) were used to create figures.

## 3. Results

### 3.1. Gut Microbiomes during Acute Murine Sepsis

To study sepsis progression in individual animals, mice either underwent cecal ligation and puncture (CLP) or served as sham controls (Figure 1A). Mice with CLP were subjected to a consecutive low-volume blood sampling (i.e., serial blood sample) and were sacrificed at preset time points (i.e., 24 h, 48 h, 72 h). Subsequently, genomic DNA of feces and peritoneal lavage as well as cfDNA were sequenced by NGS for taxonomic identification (Figure 1B). Additionally, standard microbiology was applied for species identification in terminal blood draws and peritoneal lavages (Figure 1B).

To characterize the dynamics of the mouse gut microbiome, we analyzed the gut species composition over 48 h during CLP progression. As a start, gut samples were analyzed by the whole genome shotgun sequencing at 0 h to determine the basic composition of a healthy microbiome (Figure 2A). Most mice showed *Lactobacillus reuteri* followed by *Lactobacillus johnsonii* as the most abundant species, whereas *Ruminococcus bromii*, *Bacteroides xylanisolvens,* and *Bacteroides vulgatus* typically constituted smaller but significant fractions of the microbiome. In addition, the presence of *Parabacteroides distasonis*, *Enterococcus faecalis*, *Enterococcus faecium*, *Bacteroides fragilis*, and *Candidatus Arthromitus* sp. completed obligatory fecal repertoires in all mice at 0 h. Strikingly, CLP induced significant changes in the composition of the gut microbiome at both 24 and 48 h (Figure 2B). In contrast to healthy microbiomes, species including *E. coli*, *B. vulgatus*, *E. hirae*, and *Clostridium perfringens* became predominant (for simplicity, henceforth collectively referred to as pathobiome) and outcompeted healthy gut commensals. Surprisingly, we did not observe any consistently defined post-CLP pathobiome signature during acute sepsis. In contrast, the pathobiome underwent species shifts that occurred in a highly individual manner at different time scales in each mouse (Figure 2B), which was also demonstrated by PCA analyses (Figure 2C; Appendix A). While 0 h microbiome samples formed a relatively homogenous cluster, septic samples showed a heterogeneous distribution evidenced by multidirectional variance. In addition, co-occurrence network analyses (compare [32]) did not reveal species with significant co-occurrence for a comparison of feces at 0 h with feces at 24 h.

### 3.2. Transition of the Gut Microbiome to Different Compartments

Next, we analyzed microbial colonization of the peritoneum and blood by (i) whole genome shotgun sequencing and (ii) classical microbiological culture to characterize the transition of species from the gut to systemic circulation (Figure 3). In sham mice, no pathogens were detected by culture-based methods in the peritoneal lavage and blood, while NGS identified only minor species with lowest read numbers indicative of microbial contaminants. However, pathogens in CLP mice including *Enterococcus casseliflavus*, *E. hirae*, *E. coli*, and *E. faecalis* were already found as dominant species in the blood and lavage samples by microbiological culture after 24 h (Figure 3A,B). Notably, pathogenic species colonization could differ significantly between mice and compartments, whereby NGS analysis typically revealed the same pathogens as classical culture-based microbial diagnostics (Figure 3C,D). However, additional species, including *B. fragilis*, *B. vulgatus*, *Citrobacter rodentium*, and *C. perfringens*, were exclusively identified by metagenomics sequencing. Sporadically, NGS analyses also detected species with lowest abundance (<10 normalized reads) in lavage and blood of sham mice, but those were considered to be background noise derived from contamination.

Overall, NGS significantly matched microbiological findings in septic mice for species colonizing the bloodstream (Figure 4A) and the peritoneum (Figure 4C). Venn diagrams in Figure 4B,D show that 7 out of 9 and 6 out of 9 major species detected by culture-based methods were robustly detected by NGS in the terminal blood and peritoneal lavage, respectively. Importantly, NGS allowed the detection of an additional 53 species in the blood and 114 species in the peritoneal lavage compared to classical culture-based diagnostics.

Given that gut pathobiomes constitute reservoirs for the transition of microbes to the bloodstream and/or peritoneum, we analyzed both compartments in more detail in three randomly selected individual mice over time. Data demonstrated both an idiosyncratic microbial composition for each mouse and some similar patterns across mice in the analyzed compartments (Figure 5). Feces at 0 h and 24 h post-CLP were similar for mouse 2 (Figure 5B), with only *E. coli* of significantly higher abundance. In contrast, feces at 24 h from mice 1 (Figure 5A) and 3 (Figure 5C) already show dramatic changes in comparison to the corresponding sample of the identical mice at 0 h. Additionally, the peritoneal lavage and blood samples showed different compositions between mice and compared to the corresponding fecal samples. Interestingly, both sample types (i.e., peritoneal lavage and blood) typically contained pathogenic species (e.g., *E. coli* or *E. hirae*) also identified in the fecal pathobiome at 24 h In addition, pathobiome communities in the respective samples were analyzed regarding diversity and richness (Figure 5D; Appendix A). There was no striking difference between diversity and richness for feces at 0 h and 24 h. However, lavage and terminal blood pathobiome communities were generally less rich and diverse than feces at 24 h in all three mice, also reflected by the species abundance (Figure 5A–C) where mostly one or two dominating species could be observed in each sample. In this context, co-occurrence analyses revealed no significant networks of species.

The blood pathogen identification from our CLP experiments was also retrospectively juxtaposed to readouts obtained from human patients suffering from abdominal sepsis (Figure 6; Appendix A). A qualitative comparison reveals a high interspecies similarity, given that the most prominent microbial genera detected in septic patients were also identified in CLP mice. This significant overlap of genera indicates a good correlation between human and murine pathobiome characteristics in abdominal sepsis. However, our comparison also indicated some differences in the relative occurrence of species. While *Escherichia*, *Bacteroides*, *Enterococcus*, and *Klebsiella* species demonstrated a similar occurrence, *Enterobacter*, *Lactobacillus*, and *Candida* species were more frequently detected in the murine sepsis blood samples.

### 3.3. Pathobiome Dynamics in the Blood of Septic Mice

Upon entering the bloodstream, a pathobiome can dynamically change within a short period of time (Figure 7). To qualitatively and quantitatively characterize the microbial loads in individual mice over time, we devised a workflow for absolute quantification of microbial burden in a minute amount of blood. First, we validated that pathogen identification by NGS analyses of low-volume serial blood sampling matched results from well-established, large-volume sampling (Appendix A). Subsequently, consecutive small-volume blood samples were taken at 72 h post-CLP in individual mice to characterize pathobiome dynamics during the progression of sepsis. Significantly more species were found 24 h post-CLP compared to 24 h before CLP (+4 and +3 species in mouse 1 and 2, respectively), with a rapid increase in general abundance (Figure 7A). Additionally, there was always a single species in each mouse that predominated the pathobiome: *E. hirae* in mouse 1 and *B. vulgatus* in mouse 2. Interestingly, quantitative contributions of individual species changed over time. After 48 h post-CLP, abundance of *E. hirae* and *B. vulgatus* were significantly reduced in the blood compared to 24 h. Mouse 1 had a recurring increase in pathogenic load up to 72 h and did not seem to control the infection efficiently. In contrast, mouse 2 improved over time with a decreasing microbial burden. Interestingly, the activation of the inflammatory system as well as the microbial burden appears to peak at 24 h post-CLP, as indicated by pathobiome abundance as well as by the dynamics of circulating cytokines such as IL-6 (a typical surrogate marker of systemic inflammation) (Figure 7A; Appendix A).

Our workflow for limited blood volumes also enabled time-dependent monitoring of microbial cfDNA levels from corresponding pathogenic species in single individuals over time (Figure 7B–D; Appendix A). Accordingly, we examined microbial cfDNA dynamics for the three most abundant species including *E. coli*, *B. vulgatus*, and *E. hirae* in two individual CLP mice for up to 72 h post-CLP. Compatible with previous findings (Figure 5), a high level of idiosyncrasy in the CLP response was observed. Among various patterns of dynamic changes, we also detected rapid declines for microbial cfDNAs (*E. coli* and *B. vulgatus* in mouse 2 and *E. hirae* in mouse 1), indicating a very short half-life of microbial cfDNA.

## 4. Discussion

In this study, we present a new workflow to investigate microbiomes and pathobiomes in individual mice during acute sepsis based on NGS. We observed that such pathobiomes rapidly arise at early time points (24 h) in different compartments in mice following CLP. Possibly, pathogenic species from the gut and peritoneum serve as reservoirs for the pathobiome in the blood. The transformation of a gut microbiome into a pathobiome appears to be a prerequisite for the occurrence of pathogenic species in the blood and lavage. Species frequently found in the lavage and blood at 24–48 h (i.e., *E. hirae*, *E. coli*, *B. vulgatus*, and *L. reuteri*, or *L. johnsonii*, *E. hirae*, *E. coli*, and *B. vulgatus*, respectively) were underrepresented at 0 h but were frequently detected in feces between 24 and 48 h post-CLP. These findings indicate a significant invasiveness of those species, given that the predominant overgrowth in the gut and transition into the blood and lavage takes place within a few hours during CLP. In addition to next-generation sequencing, we characterized the bacterial load in septic mice by a classical microbiological culturing. Compared to the microbial culture, NGS identified significantly more species in a timely manner. However, sequencing results must also be evaluated with respect to contaminations, as NGS is a very sensitive approach. Laboratory reagents (e.g., library preparation reagents) are not specified as pathogen-free, and minimal contamination may be detected in the final sequencing results [33,34]. Moreover, handling of samples can result in contamination by the skin microbiome species. To overcome these problems, it is mandatory to analyze negative controls and control samples of sham mice in parallel for detecting background noise. In clinical studies, sophisticated relevance scores are already used to discriminate between real signals and contamination [26].

While previous works provided some gut microbiome characteristics of CLP mice [35,36,37], to the best of our knowledge there is no direct comparison of pathobiome abundance between CLP mice and septic patients. Both mouse and human blood pathobiomes were highly idiosyncratic, indicated by a wide range of abundance of the most prominent microbial species (Appendix A). However, they displayed a large overlap in the most prominent genera. This new evidence reasserts high fidelity of the CLP model of abdominal polymicrobial sepsis also regarding pathobiome composition and dynamics. Mouse pathobiome levels in the blood have demonstrated significant dynamic changes during sepsis progression. Importantly, consecutive measurements of cfDNA levels in the blood revealed significant changes in the species-specific cfDNA abundance over time, indicating a relatively short half-life of this biomarker. Existing data estimate the half-life of human cfDNA in a range of minutes [38]. In contrast to human cfDNA, microbial cfDNA is not protected from degradation by its association with histones [39,40]. Therefore, it might be plausible that the half-life of the microbial cfDNA is even shorter than the half-life of human cfDNA. Our results indicate that species-specific cfDNAs may serve as a precise correlate of the actual microbial load in the blood, providing information about the presence/status of an infection and the efficacy of bacterial clearance (e.g., due to an antimicrobial treatment). By demonstrating successful microbiome monitoring in CLP by NGS of cfDNA, we provide an effective approach to be utilized for individualized follow-up and/or diagnostics in the clinics, as already suggested elsewhere [41].

Any individualized treatment in sepsis requires in-depth insight into its pathophysiology and comprehension of the pathobiome dynamics in an infected host. The CLP model has been widely used in preclinical research, given its many translational advantages [42,43]. We showed that even very small blood volumes are sufficient for reliable cfDNA diagnostics, indicating its robust applicability. This new applicability extends also to laboratory animal models of disease in which only small blood volumes can be collected in repetitive (survival) fashion, making investigation of the pathobiome dynamics in individual animals challenging but possible. Our findings therefore suggest that cfDNA represents a precise biomarker for pathogen identification in septic individuals in order to establish routine NGS-diagnostic workflows and to support clinical decision making in the future.

Our study has several limitations. Young female mice are not an optimal representation of typically elderly patients of different sexes who frequent suffer from various comorbidities [44] and often undergo various invasive procedures. In addition, the longitudinal follow-up is available in only a few mice (mostly 2–3 per group), and not each time point was examined per mouse due to blood volume sampling restrictions and the pioneering character of the study. This analytical work must be further expanded in a separate study to provide more comprehensive insight into those protracted and heterogeneous dynamics.

## 5. Conclusions

The abovementioned points indicate the need to investigate sepsis progression on an individual level, especially with respect to future use in clinics. It is expected that general species dominance or microbiome composition in one compartment does not allow conclusions to be made about complex sepsis development in a patient. Hence, individual and continuous monitoring must be undertaken to obtain diagnostic information that is needed for correct treatment. However, further validation in human patients is needed to overcome the limitations of the mouse model.

## Figures and Tables

**Figure 1 microorganisms-11-00627-f001:**
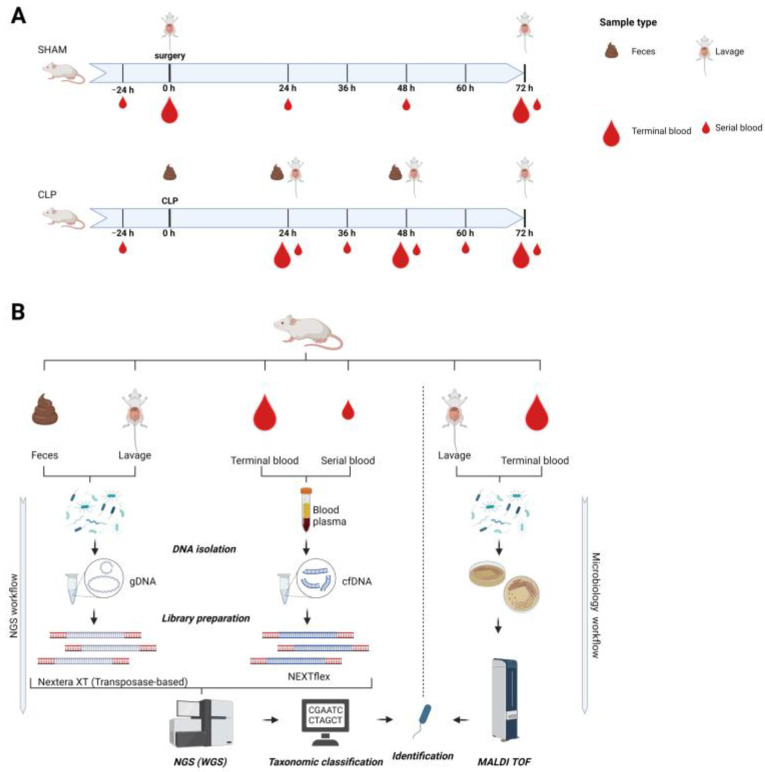
Schematic illustration of the study design (**A**) and of the analytical workflow (**B**). (**A**): Three-month-old female mice (*n* = 45) were subjected to polymicrobial cecal ligation and puncture (CLP) sepsis (lower axis) or served as sham controls (upper axis) and were sampled in defined intervals for blood, feces, and abdominal lavage until 72 h post-CLP. Small blood drop indicates a serial small-volume sampling; large blood drop indicates terminal blood sampling. Short-interval serial blood sampling was performed maximally three times per individual mouse (i.e., serial blood sampling time points varied among mice to cover intervals smaller than 24 h) followed by terminal large-volume sampling. In each mouse, feces were collected at CLP and at the sacrifice time point. In each mouse, abdominal lavage was performed at the sacrifice time point. Body weight and temperature were taken daily. (**B**): NGS workflow was performed for all sample types. DNA was isolated, and all libraries were used for whole genome shotgun sequencing; nonmurine reads were classified and used for identification of species. Abdominal lavage and terminal blood were used for microbiological culturing in parallel. Growing cultures were used for species identification by means of MALDI TOF.

**Figure 2 microorganisms-11-00627-f002:**
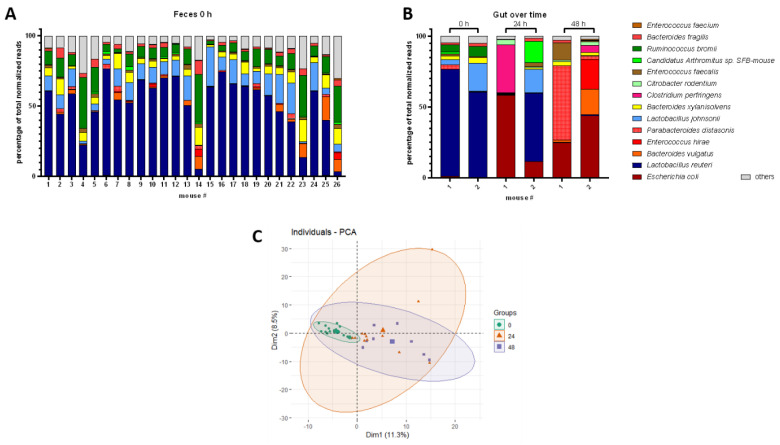
Healthy and post-CLP gut microbiome. (**A**): Healthy gut microbiome (0 h), percentage share of total normalized sequencing reads of top species shown for different mice. (**B**): Septic gut microbiome at 0 h, 24 h, and 48 h post-CLP, percentage share of total normalized sequencing reads of top species shown for different mice. (**C**): PCA of microbiomes at 0 h, 24 h, and 48 h post-CLP. Time points are indicated by color and shape of the symbols; bold symbols are centroids of the corresponding group.

**Figure 3 microorganisms-11-00627-f003:**
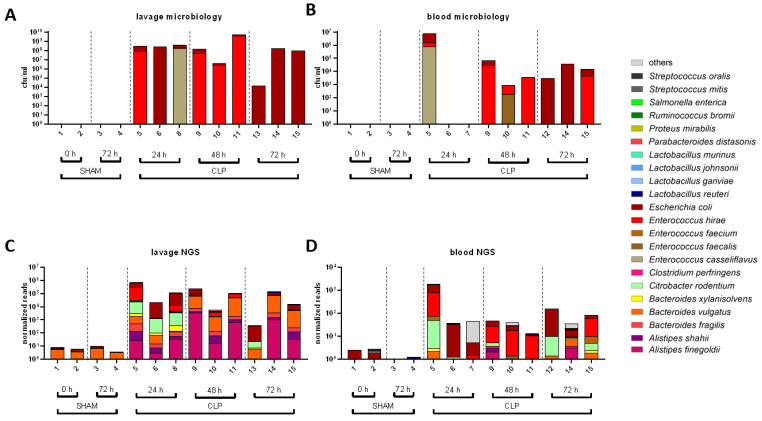
Transition of microbes into peritoneum and blood as revealed by NGS and microbiology. (**A**–**D**): The most frequent bacterial species found in the lavage (**A**,**C**) and blood (**B**,**D**) detected by the culture-based methods (**A**,**B**) and by whole genome shotgun sequencing (**C**,**D**).

**Figure 4 microorganisms-11-00627-f004:**
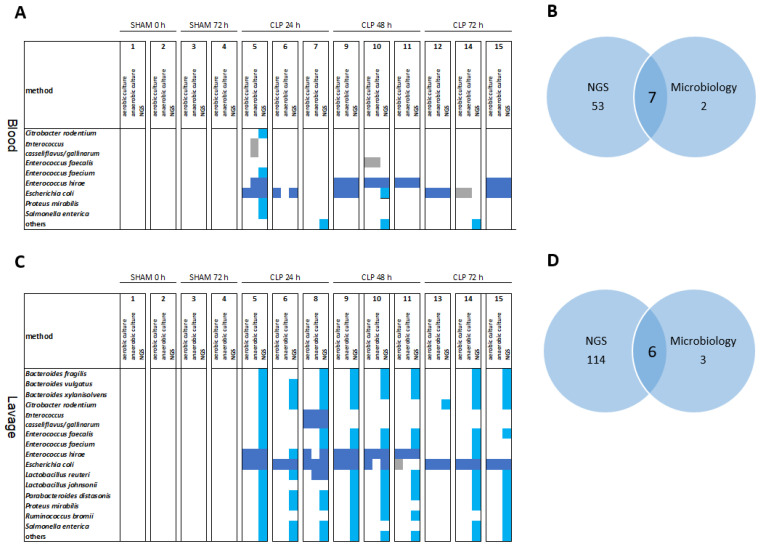
Overlap of microbes in the peritoneum and blood as revealed by NGS and microbiology. The most abundant species are shown for 0–72 h in sham and CLP mice. Heat maps (**A**,**C**) and Venn diagrams (**B**,**D**) of the most abundant species in the blood (**A**,**B**) and abdominal lavage (**C**,**D**) over 0–72 h in sham and CLP mice. Light blue squares, species detected by NGS (with >10 normalized reads); dark blue squares, species detected by NGS and either aerobic or anaerobic culture; gray squares, species detected by aerobic and/or anaerobic culture only.

**Figure 5 microorganisms-11-00627-f005:**
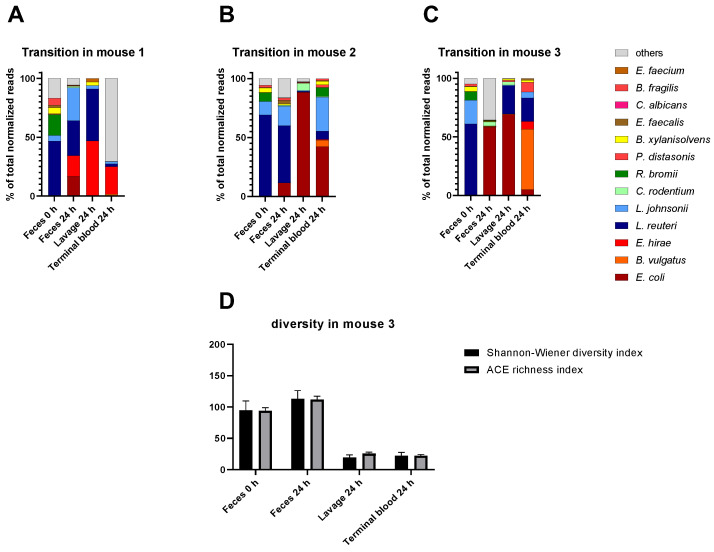
Transitions of species from the feces to bloodstream/peritoneum at 24 h post-CLP in three mice. (**A**–**C**): Percentage of the most abundant species in feces at 0 h and 24 h, lavage at 24 h, and terminal blood sample at 24 h after CLP. (**D**): Community diversity and richness shown for all sample types of mouse 3 indicated by Shannon–Wiener diversity index and ACE richness index. Standard errors are shown.

**Figure 6 microorganisms-11-00627-f006:**
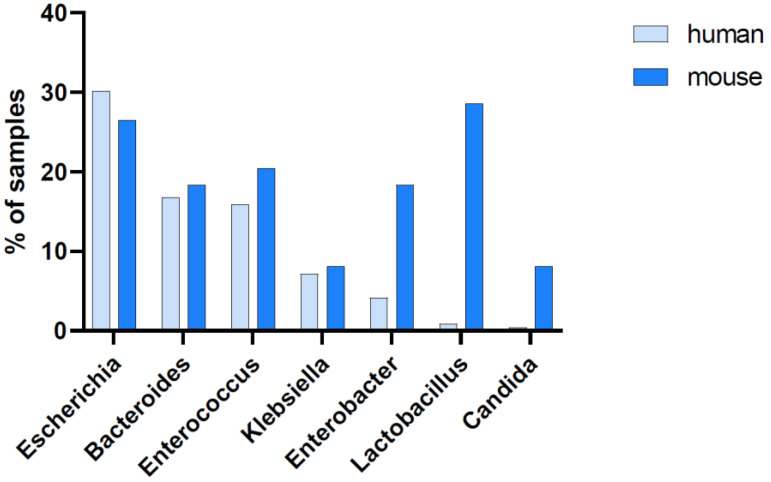
Abundance of the most prominent genera in the blood. Blood samples collected from septic humans (*n* = 239 samples of 48 individuals at different time points) and mice (*n* = 49 terminal blood samples of 49 individuals). The seven most prominent genera found in humans and mice are shown; their abundance is indicated by the percentage of blood samples showing >10 normalized sequencing reads for the respective genus out of all available blood samples.

**Figure 7 microorganisms-11-00627-f007:**
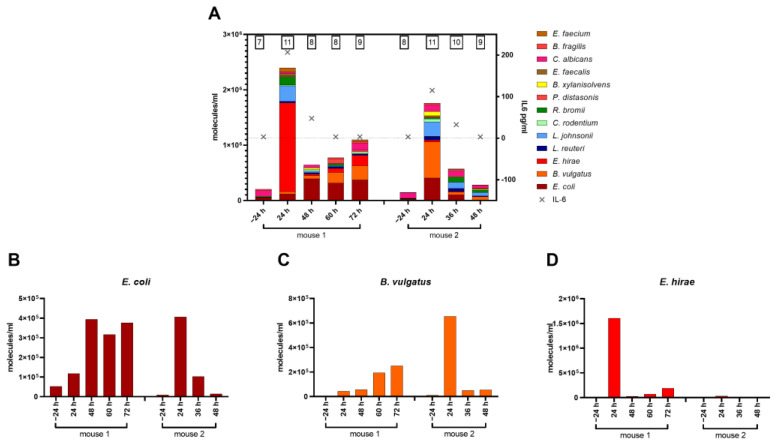
(**A**): Concentration for top species shown in molecules/mL of serial blood samples for two mice at different time points. The numbers above bars indicate the number of detected top species. Crosses indicate a corresponding IL-6 concentration in the blood per time point. (**B**–**D**): Time course of the most abundant species in molecules/mL of serial blood samples in two individuals.

## Data Availability

The datasets generated and/or analyzed during the current study are openly available in the Sequence Read Archive (SRA) of the NIH, under the following link: https://www.ncbi.nlm.nih.gov/bioproject/PRJNA867529, accessed on 27 February 2023.

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
