# Peer review of "From Gut to Blood: Spatial and Temporal Pathobiome Dynamics during Acute Abdominal Murine Sepsis"

_microorganisms, 2023, doi:10.3390/microorganisms11030627_

Round 1
Reviewer 1 Report
The authors established a method and its biomarkers to reliably study the emergence of pathobiomes and to monitor their respective dynamics. They found that pathobiomes serve as reservoirs following CLP for the transition of pathogens into the bloodstream, and cfDNA can serve as a precise biomarker for pathogen identification in blood. This study is interesting and important. There a few points that can be improved:
1. It is recommended to analyze potential relationship between alpha-diversity index (including community diversity, richness, and evenness) and the pathogen infection, like doi: 10.1128/Spectrum.00229-21.
2. It is recommended to analyze the co-occurence network among pathogenic taxa of pathobiomes from the infected versus the non-infection, like: 10.1016/j.biortech.2022.127549.
Author Response
- It is recommended to analyze potential relationship between alpha-diversity index (including community diversity, richness, and evenness) and the pathogen infection, like doi: 10.1128/Spectrum.00229-21.
We analyzed the diversity and richness for the samples of the three mice shown in main Figure 5 as well as samples at different time points for each compartment. A panel 5D was added to main Figure 5 and an additional file (supplementary figure S3) was created.
- It is recommended to analyze the co-occurence network among pathogenic taxa of pathobiomes from the infected versus the non-infection, like: 10.1016/j.biortech.2022.127549.
We analyzed co-occurrence networks according to the recommended paper. Baseline was compared to end time points after CLP: feces 0 h vs. 24 h, lavage of sham mice vs. lavage of CLP mice at 72 h and terminal blood of sham mice vs. terminal blood of CLP mice at 72 h. However, no network with significant co-occurrences formed so we could not create a meaningful figure. Nevertheless, we included a sentence next to Figure 2 and Figure 5 in the manuscript pointing to the fact that network analyses were performed.
Reviewer 2 Report
In this study, the authors developed a customized analytical workflow for the mouse cecal ligation and puncture (CLP) model of abdominal sepsis to characterize individualized microbial dynamics of the gut, blood, and abdominal compartments before and during the acute phase of polymicrobial sepsis. Additionally, the authors also compared the blood pathobiome of septic mice with that of human patients with abdominal sepsis. The study provides an interesting insight into the dynamics of pathobiomes in sepsis, however, it may be beneficial for the authors to revise the manuscript to improve its clarity and readability.
1. In the final paragraph of the introduction, the authors should briefly introduce the significance of this study.
2. It is suggested that the authors provide the total number of experimental animals used in the study, as well as the specific number of animals per group, in the Materials and Methods section.
3. The authors have provided an ethics agreement related to the use of experimental animals in the study. It is recommended that they also provide an ethics agreement related to the use of clinical patients in the study.
4. It is recommended that the authors include a statistical section in which they introduce the specific statistical methods used in the analysis of the data.
5. The authors should explain why they did not use elderly mice in their experiments, as this would better simulate real-world clinical scenarios.
Author Response
- In the final paragraph of the introduction, the authors should briefly introduce the significance of this study.
See additional sentence in the manuscript.
- It is suggested that the authors provide the total number of experimental animals used in the study, as well as the specific number of animals per group, in the Materials and Methods section.
See additions in the Materials and Methods section.
- The authors have provided an ethics agreement related to the use of experimental animals in the study. It is recommended that they also provide an ethics agreement related to the use of clinical patients in the study.
The Ethics Committee of the Medical Faculty of Heidelberg approved the clinical study of Grumaz et al., 2019 (doi:10.1097/CCM.0000000000003658) with the Trial Code No. S-097/2013. Data of that study was re-analyzed in our manuscript. The information was added to the Materials and Methods section.
- It is recommended that the authors include a statistical section in which they introduce the specific statistical methods used in the analysis of the data.
In our case, we report a proof of concept study in which only few mice were analyzed. In addition, statistical analyses on the highly variable data of individual mice did not add any valuable information about species occurrence/pathobiome dynamics. Hence, no statistical analyses were performed in the manuscript that need to be reported in the methods section.
- The authors should explain why they did not use elderly mice in their experiments, as this would better simulate real-world clinical scenarios.
Due to the 3R principles and practical reasons, we chose not to use elderly mice. We wanted to minimize animal use and aged mice have an increased mortality/morbidity, thus, greater counts would have been needed. To keep the n as small as possible we used 3 month-old mice.
Given that this was a proof of concept study, mature mice are an acceptable model. In fact, from the analytical perspective this could have been more challenging (given the aged mice combat such infections less effectively).
Proving the efficacy of our analytical workflow in middle-aged mice very much elevates its reliability as presence of pathogens in aged mice is expected to be greater (due to impaired microbial clearance). In future studies, we will also involve aged septic mice to confirm our current results and investigate potential differences. This, however, needs to be designed as a separate study.
Reviewer 3 Report
The study by Hartwig C et al, titled “From gut to blood: spatial and temporal pathobiome dynamics during acute abdominal murine sepsis” aims to investigate the transition of microorganisms from the gut to the peritoneum and bloodstream by NGS and microbial culture. Although it is an interesting article, I have some concerns about this article.
Specific comments
Authors have only female mice for CLP. Is there any sex specific effect?
It would be nice to mention the animal body weight in method section.
There is no detail for cytokine measurement in the method section.
Figure 5 they have used 3 mice for transition and 2 mice in figure 7. Is this statistically significant? Author should increase animal number for experiments.
Author should provide figure legend for all supplementary figures and table.
It would be nice to combine figure 3 and 4.
It would be interesting to check the organs (liver and lungs) histology in respective timepoint.
Author Response
- Authors have only female mice for CLP. Is there any sex specific effect?
Given that this was a first ever proof-of-concept study of that type, we aimed to avoid any additional confounding factors like sex, age, co-morbidities and chose middle-aged female mice to observe the dynamics induced by CLP. This was also dictated by the NIH recommendation on using female sex in pre-clinical studies (to counterbalance an overrepresentation of male sex).
This choice had also a practical aspect: males have to be kept separately and females 5/cage. This makes our findings regarding individual pathobiome diversity even more interesting.
Sex-specific effects (similar to an advanced age as another component) require a separate attention and should be addressed in a follow-up study.
- It would be nice to mention the animal body weight in method section.
A paragraph was added to the methods section.
- There is no detail for cytokine measurement in the method section.
All cytokines were measured by a highly sensitive automatized Luminex bead-based technology. A method description for the cytokine measurements was added to the methods section.
- Figure 5 they have used 3 mice for transition and 2 mice in figure 7. Is this statistically significant? Author should increase animal number for experiments.
Due to the proof of concept character of the study, there is no statistical analysis in the two figures (and such a comparison was not intended). We simply wanted to establish whether such an approach is at all technically feasible. In a follow-up study, animal numbers will be increased.
- Author should provide figure legend for all supplementary figures and table.
Figure legends are added to the “Supplementary Materials” section.
- It would be nice to combine figure 3 and 4.
If the editors specifically request it, we can certainly combine figures 3 and 4 to create a large, multi-panel figure. However, after testing this possibility we concluded that these data are presented more clearly if separated.
- It would be interesting to check the organs (liver and lungs) histology in respective time point.
Unfortunately, we did not sample liver and lung tissue as it was out of scope for this study. The design of our study was already very challenging and we elected to simplify it to reduce possibility of experimental errors. We fully agree that these organs are certainly a very interesting analytical target for a follow-up study.
Round 2
Reviewer 1 Report
The manuscript has been sufficiently improved to warrant publication in Microorganisms.
Reviewer 3 Report
The manuscript has been revised and improved accordingly to the reviewers comments.